# Dopamine Function and Hypothalamic-Pituitary-Thyroid Axis Activity in Major Depressed Patients with Suicidal Behavior

**DOI:** 10.3390/brainsci12050621

**Published:** 2022-05-10

**Authors:** Fabrice Duval, Marie-Claude Mokrani, Vlad Danila, Alexis Erb, Felix Gonzalez Lopera, Mihaela Tomsa

**Affiliations:** Pôle 8/9-APF2R, Centre Hospitalier, 68250 Rouffach, France; marieclaudejean2010@hotmail.fr (M.-C.M.); v.danila@ch-rouffach.fr (V.D.); a.erb@ch-rouffach.fr (A.E.); f.gonzalez@ch-rouffach.fr (F.G.L.); m.tomsa@ch-rouffach.fr (M.T.)

**Keywords:** depression, suicidal behavior disorder, thyrotropin-releasing hormone (TRH) test, thyrotropin, dopamine, apomorphine

## Abstract

Involvement of the dopaminergic (DA) and hypothalamic-pituitary-thyroid (HPT) systems in suicidal behavior is still poorly understood. We assessed multihormonal responses to apomorphine (APO; a short acting DA receptor agonist) and 8 AM and 11 PM protirelin (TRH) tests in 30 medication-free DSM-5 euthyroid major depressed inpatients with suicidal behavior disorder (SBD) (current, *n* = 14; in early remission, *n* = 16) and 18 healthy hospitalized control subjects (HCs). Compared to HCs, responses to APO and TRH tests were unaltered in SBDs in early remission. However, current SBDs exhibited increased APO-induced growth hormone (GH) and adrenocorticotropin (ACTH) stimulation, and reduced 11 PM thyrotropin (TSH) and ∆∆TSH values (difference between 11 PM and 8 AM TRH-TSH responses). In current SBDs, the association between high APO-GH concentrations and low ∆∆TSH values was more common in recent suicide attempters than in past suicide attempters. These preliminary results suggest that co-occurring alterations in the DA and HPT systems (i.e., DA receptor hyperresponsiveness associated with decreased hypothalamic TRH drive) may contribute to the pathophysiology of suicidal behavior. Conversely, normalization of DA and TRH functions might reflect a process of recovery from suicidality. Thus, our findings suggest that drugs targeting the DAergic and TRH systems could be relevant in suicide prevention.

## 1. Introduction

Over several decades, research on suicidal behaviors in major depression has focused on abnormalities of serotonergic (5-HT) and noradrenergic (NA) neurotransmission as well as those of the hypothalamic-pituitary-adrenal axis (HPA) [1,2]. However, there is growing evidence that the hypothalamic-pituitary-thyroid (HPT) axis [3,4,5] and the dopamine (DA) system [6,7] are altered as well.

On the one hand, some, but not all, studies indicate that suicide attempts in depressed patients may be associated with reduced central DAergic neurotransmission. A recent meta-analysis revealed that suicide attempters have lower levels of homovanillic acid (HVA)—a DA metabolite—in the cerebrospinal fluid (CSF) than non-attempters [8]. With regard to DA receptors, results are inconsistent: increased, decreased, or unchanged sensitivity or binding have been found in depressed patients with suicidal behavior [6,7,9,10,11,12,13,14].

Recently, Pettoruso et al. [15] reported that striatal DA transporter (DAT) availability was reduced in depressed patients with high suicidality compared to those at low risk of suicide; interestingly, decreased DAT density may reflect a compensatory down-regulation secondary to blunted DA signaling within mesolimbic pathways [16].

On the other hand, disturbances of the circadian rhythm of the HPT axis have been consistently found in acute euthyroid depressed patients, such as a failure of the normal nocturnal surge of thyrotropin (TSH) associated with a blunted 11 PM TSH response to protirelin (TRH) test, and a reduced difference between 11 PM and 8 AM TRH-TSH responses (∆∆TSH) (for a review see [17]). Whereas studies on TSH response to morning TRH challenge test have yielded discrepant findings in depressed patients with suicidal behavior (reviewed in [4]), our group reported altered TSH responses to TRH at 11 PM, and consequently reduced ∆∆TSH values, in recent attempters [4,18]. Interestingly, this chronobiological abnormality was not observed in depressed patients with a history of suicidal behavior but who had not attempted suicide for more than a year [4]. From a pathophysiological viewpoint, a deficient TRH-TSH response indicates a reduced TSH pituitary reserve and/or decreased functionality of the pituitary TRH receptors; for chronobiological reasons, TRH receptor hyposensitivity is more marked in the evening than in the morning [19]. According to the clinical picture, central TRH secretion might be increased, normal or decreased, regardless of HPA axis activity [17,20]. In depressed patients without a suicidal history, TRH receptor downregulation may be induced by endogenous central TRH oversecretion [3] and may lead to understimulation of TSH synthesis. In recent suicide attempters, blunted TRH-TSH responses associated with decreased free thyroxine (FT4) levels [4,21,22] might reflect decreased central TRH activity. Indeed, in a post-mortem study [23] carried out in depressed patients with persistent suicidal thoughts, a decrease in TRH mRNA levels in the paraventricular nucleus was reported, which is consistent with reduced hypothalamic TRH secretion. Given the role of endogenous TRH as a homeostatic neuromodulator acting on monoaminergic neurotransmission [24], we previously discussed that the TRH-mediated compensatory mechanism (i.e., increased central TRH secretion) is triggered in order to correct reduced 5-HT activity in non-suicidal depressed patients [5,18,25]. In depressed patients with a history of suicidal behavior—where 5-HT hypoactivity is supposed to be a “trait marker” [1,26]—this homeostatic compensatory mechanism is not elicited. Depending on the time elapsed since the last suicide attempt, we recently suggested that TRH secretion is either reduced (when the last suicide attempt occurred within the past year) or normal (when the last suicide attempt occurred 12 to 24 months prior to evaluation) [4].

Based on human and animal data, it has been shown that the DA and HPT systems are closely interrelated. Indeed, DA-D2 receptors stimulate the release of TRH at the hypothalamic level [27] and inhibit pituitary TSH production [28,29], and TRH stimulates DA synthesis and release in several brain regions [30,31,32]. In psychiatric patients, DA function can be evaluated by apomorphine (APO)-mediated hormonal responses [33]. APO is a non-selective DA receptor agonist that inhibits prolactin secretion (via D2 receptors in the pituitary) and stimulates (via D1-like and D2-like receptors in the hypothalamus) the release of growth hormone releasing hormone (GHRH) and corticotropin-releasing hormone (CRH), which then stimulate growth hormone (GH), and adrenocorticotropic hormone (ACTH) and cortisol (COR), respectively [34]. In depressed inpatients without a history of suicidal behavior, TRH and APO test responses are inversely correlated [5]. Furthermore, those with reduced ∆∆TSH values (consistent with increased hypothalamic TRH drive leading to decreased TRH receptor chronesthesy on pituitary thyrotrophs) show normal hormonal responses to APO, while those without an HPT axis abnormality (i.e., with normal ∆∆TSH values) exhibit markedly blunted GH and ACTH/COR responses to APO (suggesting decreased hypothalamic DA-receptor function). Thus, we hypothesized that in non-suicidal depressed patients, the TRH-mediated compensatory mechanism is triggered when the hypothalamic DA system is efficient [5].

The aim of this present study was to clarify the relationships between the DA and HPT systems in depressed patients who committed a suicide attempt within the past two years. We hypothesized that the co-occurrence of alterations in the DA and HPT systems could contribute to the pathophysiology of suicidal behavior.

## 2. Materials and Methods

### 2.1. Subjects

This retrospective study was conducted in accordance with the Declaration of Helsinki and approved by the local ethics committee (Rouffach Hospital Review Board). Thirty patients (aged 20 to 49 years) and 18 healthy hospitalized volunteers (aged 22 to 54 years) participated in this study. All enrolled patients were hospitalized for an acute exacerbation of their depressive symptoms or admitted for suicidal ideation or attempts. All met DSM-5 criteria for major depressive disorder (MDD) [35] and suicidal behavior disorder (SBD) [36]; the essential manifestation of the latter is a suicide attempt committed within the last 24 months. The diagnosis was made by two experienced psychiatrists blind to neuroendocrine findings. All patients had a history of recurrent major depression without psychotic features. According to the proposed DSM-5 criteria, when the suicidal behavior occurred 12 to 24 months prior to assessment, patients were classified as “SBD in early remission” (*n* = 16); when the suicide attempt occurred within the past year, patients were classified as “current SBD” (*n* = 14). Current SBDs were then classified as recent suicide attempters (*n* = 7; 4 male/3 female; mean age ± SD, 32.4 ± 5.2 years) if the suicidal act occurred within the past two months prior to evaluation (neuroendocrine tests were performed at least 3 weeks after the most recent suicide attempt), or past suicide attempters (*n* = 7; 3 male/4 female; mean age ± SD, 33.6 ± 8.6 years) if the most recent suicide attempt occurred 3 to 11 months prior to evaluation. Among current SBDs, 5 used a violent method (i.e., hanging, jumping from a height) to commit their latest suicide attempt and the remaining 9 used non-violent methods (i.e., superficial wrist cuts, drug overdose with selective reuptake inhibitors/noradrenergic and serotonergic antidepressants, anxiolytics/hypnotics, possibly associated with alcohol). To be included, patients had to have an initial score ≥ 18 on the 17-item Hamilton Depression Rating Scale (HAM-D) [37]. All patients gave their informed consent prior to participation. All had been drug-free for at least 10 days (wash-out was supervised in hospital), as all were treated with antidepressants at the time of hospital admission. It must also be noted that 10 patients from the final sample (i.e., 33%) had participated in our previous study [4].

In order to assess the level of activity for DAergic function and the HPT and HPA axes in the patients, we selected data for 18 hospitalized healthy control subjects (HCs) from our database. None of them had a personal or family history of major psychiatric illness (as evaluated by means of the SADS-LV and the Research Diagnostic Criteria—Family History [38]), and all were without current medication use. HCs were hospitalized 3 days before testing and remained hospitalized throughout the testing period (4 days).

Routine physical examination and laboratory tests were performed in all subjects. All subjects had basal TSH, free thyroxine (FT_4_), free triiodothyronine (FT_3_), ACTH, COR and GH values within the normal range. The body mass indexes for the participants needed to be between 18.5–25 kg/m^2^ (i.e., normal weight). We excluded subjects with a personal history of endocrinopathy; drug abuse (including alcoholism); and previous treatment with monoamine oxidase inhibitors, lithium salts, carbamazepine, long-acting antipsychotics, fluoxetine (or other antidepressants with a very long half-life of elimination), and/or electroconvulsive therapy. All women were free from hormonal contraception and were tested outside the periovulatory period. Patients and controls were on a caffeine-restricted diet for at least three days prior to neuroendocrine testing, and their environment was synchronized, with diurnal activity from 8 AM to 11PM.

### 2.2. Procedures

We measured TSH, FT4, and FT3 levels before and after TRH given at 8 AM and 11 PM on the same day (day 1), using 200 µg of synthetic TRH IV [19] (TRH Ferring^®^, Ferring Pharmaceuticals, Kiel, Germany). The rationale for this procedure has been discussed elsewhere [19]. After an overnight fast, subjects were awoken at 7 AM. An indwelling cannula was inserted into a forearm vein and kept open with an isotonic saline infusion. The first TRH stimulation test was carried out at 8 AM, and blood samples were taken 15 min before TRH injection, immediately before TRH injection, and 15, 30 and 60 min after TRH injection. The TRH test was repeated at 11 PM, on the same day, using the same procedure; subjects were awake before and during the sampling and kept without food from 6 PM. To assess the possible influence of hypercortisolemia on TRH and APO test responses, a dexamethasone (DEX) suppression test (DST) was started at midnight (after the second TRH test) with oral ingestion of 1 mg of DEX (Dectancyl, Laboratoires Roussel, Paris, France), followed by blood samples drawn for the assay of serum COR at 8 AM, 4 PM and 11 PM the next day (day 2) [39].

On day 4, an APO test (SC injection of 0.75 mg Apokinon, Laboratoires Aguettant, Lyon, France) [34] was carried out at 9 AM after an overnight fasting. Blood was drawn at −30, −15 and 0 min before APO administration and further samples were collected at 15, 30, 60, 90, 120 and 150 min for the assay of GH, ACTH and COR (following APO). Throughout the test, subjects were in bed and did not smoke.

### 2.3. Assays

Blood samples were immediately centrifuged at 3000 rpm and 4 °C, and the sera were then stored at −20 °C until assay. All hormonal concentrations were determined by immunoassay techniques based on enhanced luminescence. Average intra-assay and inter-assay coefficients of variation were as follows: TSH, 3.4–4.8%, sensitivity < 0.01 mU/L (Access Hypersensitive hTSH Assay, Beckman Coulter, Inc., Fullerton, CA, USA); FT4, 2.8–5.1%, sensitivity < 3.2 pmol/L (Access Free T4 Assay, same supplier); FT3, 4.7–4.8%, sensitivity < 1.4 pmol/L (Access Free T3 Assay, same supplier); COR, 5.1–6.8%, sensitivity < 11 nmol/L (Access Cortisol Assay, same supplier); ACTH, 2.7–7.9%, sensitivity < 1 ng/L (Nichols Advantage^®^ ACTH, Nichols Institute Diagnostics, San Juan Capistrano, CA, USA); GH assay, 3.9–7.5%, sensitivity < 0.1 µg/L (Nichols Advantage^®^ hGH, same supplier).

### 2.4. Data Analysis

Baseline TSH (TSHBL), FT4 (FT4BL) and FT3 (FT3BL) values were defined as the mean of the two samples before TRH injection. The TSH response to TRH (∆TSH) was defined as the maximum increment after TRH injection above TSHBL level; ∆∆TSH was defined as the difference between 11 PM-∆TSH and 8 AM-∆TSH values [19]. Post-DST COR suppression was evaluated as the maximum COR level in any blood sample obtained at 8 AM, 4 PM, and 11 PM on day 2 [39]. Baseline GH value (GHBL) before APO was obtained by averaging GH values from time points −30, −15 and 0 min. To be included in this study, subjects had to have, before APO, a GHBL value < 2 µg/L. Since ACTH and COR concentrations decrease markedly in the morning—owing to the normal circadian rhythm—we used the ACTH and COR values at 0 min (i.e., immediately before injection of APO) as the baseline values (ACTHBL, CORBL). GH, ACTH and COR responses were measured by subtracting the baseline level from the peak level after APO (i.e., ∆GH, ∆ACTH and ∆COR, respectively) [34].

### 2.5. Statistical Analyses

Statistical analyses were performed using software from the R Project for Statistical Computing [40]. We employed nonparametric statistical tests because some data were not normally distributed (according to the Kolmogorov–Smirnov test). The comparisons between different SBD groups and the control group were tested with the Mann–Whitney two-tailed test (*U*-test); formal corrections for multiple comparisons were not needed because we made planned comparisons. Relationships between quantitative data were estimated with Spearman’s rho (ρ) statistic. Optimal cut-off values were determined using receiver operating characteristic (ROC) curve analysis [41]; qualitative data were analyzed using Fisher’s exact test (two-tailed). Logistic regression was performed for binary dependent variables. Results were considered significant when *p* ≤ 0.05.

## 3. Results

### 3.1. Comparison between Healthy Control Subjects and Depressed Patients with Suicidal Behavior Subgrouped according to DSM-5 Specifiers (i.e., Current or in Early Remission)

The three subject groups were comparable for age and gender distribution (Table 1).

#### 3.1.1. Apomorphine Test Results

Hormonal responses to APO were independent of BL levels and gender. Although APO-ACTH and COR responses (expressed as ∆) were unrelated to age, GH responses (∆GH) were negatively correlated with age, both in HCs (ρ = −0.77; *n* = 18; *p* = 0.0002) and SBDs (ρ = −0.40; *n* = 30; *p* < 0.03); however, in each SBD group, this correlation was not significant (SBDs in early remission, ρ = −0.46, *n* = 16, *p* = 0.07; current SBDs, ρ = −0.14, *n* = 14, *p* = 0.63). Pre- and post-APO COR values were not significantly different between SBDs and HCs. However, current SBDs showed higher ∆GH and ∆ACTH values than SBDs in early remission and HCs (Table 1). Univariate binary logistic regression confirmed that increasing ∆ACTH and ∆GH values were more likely associated with the “current” specifier for SBD (Table 2).

Owing to the wide variation in ∆ACTH values, no threshold for an exaggerated response could be defined. When using a ∆GH value of more than 20 µg/L to define an increased response, seven current SBDs (50%), two SBDs in early emission (12%) and two HCs (11%) showed high responses (Figure 1a). Increased ∆GH values were more frequent in current SBDs than in SBDs in early remission and HCs (*p* = 0.04 and *p* = 0.02, respectively, by Fisher’s exact test). Multivariate logistic regression analysis after adjusting for potential confounders—age (by decade), gender and severity of depressed symptoms (moderate, 18 ≤ HDRS score ≤ 23; severe, ≥24 [42])—revealed that high ∆GH values (i.e., > 20 µg/L) were independently associated with the current SBD category (OR, 8.90; 95% CI, 1.18–67.15; *p* = 0.03).

For each SBD group, depression severity (as evaluated with the HAM-D scores) or subtype (i.e., with or without melancholic features) and number of suicide attempts did not influence hormonal responses to APO. Additionally, in the current SBD group, hormonal responses to APO were not significantly different between recent (*n* = 7) and past (*n* = 7) attempters, and between violent (*n* = 5) and non-violent (*n* = 9) suicide attempters. However, there were negative relationships between the time elapsed since the last suicide attempt and ∆ACTH and ∆GH values (Figure 2); regarding ∆COR, this relationship was not significant (ρ = −0.30, *n* = 30, *p* = 0.12).

#### 3.1.2. TRH 8 AM and 11 PM Test Results

Confirming our previous study [4], FT4BL, FT3BL and TSH (pre- and post-TRH) levels were comparable between SBD patients in early remission and HCs (Table 1). Compared to HCs, current SBDs showed lower FT4BL (at 8 AM) and TSHBL (at 11 PM) values, lower 11 PM-TSH responses to TRH (11 PM-∆TSH) and markedly decreased ∆∆TSH values. Compared to SBDs in early remission, 11 PM-∆TSH and ∆∆TSH values were reduced in current SBDs, whereas FT4BL and 11 PM-TSHBL values only tended to be lowered (*p* = 0.06 and *p* = 0.08, respectively). TSH values at 8 AM (pre- and post-TRH) and FT3BL did not distinguish current SBDs from SBDs in early remission and HCs.

As illustrated in Figure 1c, the percentage of reduced ∆∆TSH values (i.e., ≤2.5 mU/L [25]) was higher in current SBDs (79%, *n* = 11) than SBDs in early remission (19%, *n* = 3; *p* < 0.001, by Fisher’s exact test) and HCs (5%, *n* = 1; *p* < 0.00003, by Fisher’s exact test). In current SBDs, thyroid hormone values (FT4BL, FT3BL, TSHBL and TRH-TSH responses) were comparable between violent and non-violent suicide attempters, and between recent and past suicide attempters.

Logistic regression analysis confirmed that decreasing FT4BL and 11 PM-TSH values (pre-and post-TRH, and ∆∆TSH) were more likely associated with current SBD status (Table 2). In addition, current SBDs were 15.9 times more likely to exhibit low ∆∆TSH values (i.e., ≤2.5 mU/L) than SBDs in early remission.

In our population, thyroid hormone levels were not significantly influenced by age and gender. In the SBD groups, the HPT parameters were not related to severity of the illness, subtype of depression and number of suicide attempts. However, among SBDs as a group, there was a relationship between the time elapsed since the last suicide attempt and ∆∆TSH values (Figure 2); however, this relationship was not significant for 8 AM-∆TSH and 11 PM-∆TSH values (ρ = 0.26 and ρ = 0.35, respectively).

#### 3.1.3. Dexamethasone Suppression Test

Post-DST COR suppression was comparable across the three diagnostic groups. The incidence of nonsuppression of COR after DEX (i.e., highest post-DST COR level > 130 nmol/L [43]) was observed in three current SBDs (21%), four SBDs in early remission (25%) and one HC (5%).

The DST results were independent of age, gender, severity or subtype of depression and number of suicide attempts.

### 3.2. Cross Correlations between Hormonal Responses to Apomorphine and TRH Tests in Patients and Controls

As shown in Table 3, hormonal responses to APO (i.e., ∆GH, ∆ACTH and ∆COR) were related (albeit not always significantly). Furthermore, 8 AM-∆TSH and 11 PM-∆TSH values were highly correlated in each diagnostic group. The values for ∆∆TSH were related to 11 PM-∆TSH values (except in SBDs in early remission) but not 8 AM-∆TSH values. In the overall population and in each diagnostic group, the HPA axis activity, as evaluated by the DST, did not influence APO and TRH test responses. Interestingly, there were no meaningful correlations between TRH and APO test responses in the SBD and HC groups.

### 3.3. Distribution of ∆GH and ∆∆TSH Status among Patients according to the Time Elapsed since the Last Suicide Attempt

In current SBDs, high ∆GH (>20 µg/L) and low ∆∆TSH (≤2.5 mU/L) values co-occurred in five recent suicide attempters (71%) and none past attempters (*p* = 0.02 by Fisher’s exact test). This co-occurrence was also more frequent in recent suicide attempters than SBDs in remission (*n* = 1; *p* < 0.003 by Fisher’s exact test).

Moreover, the association between normal ∆GH and low ∆∆TSH values was more frequent in current SBDs with a past suicide attempt (*n* = 5; 71%) than in SBDs in remission (*n* = 2; *p* = 0.01 by Fisher’s exact test).

## 4. Discussion

This study demonstrates that non-psychotic MDDs meeting the proposed research DSM-5 criteria for suicidal behavior disorder (SBD) exhibit various degrees of dysregulation of the DAergic and thyrotropic systems. SBDs in early remission show comparable APO and TRH test responses to those of HCs, whereas current SBDs exhibit increased APO-induced ACTH and GH stimulation and altered HPT axis activity, as reflected by lowered FT4BL (still within the normal reference range) and reduced evening TSH values (at baseline and after TRH stimulation, and consequently markedly decreased ∆∆TSH values). In our population, changes in hormonal responses to APO and TRH tests do not appear to be significantly influenced by the severity or subtype of depression, number of suicide attempts, violence of method or HPA axis activity (as evaluated by post-DST cortisol levels). Unlike depressed inpatients without a history of suicidal behavior [5], APO and TRH test responses were uncorrelated in SBDs, suggesting there was no link between DA and HPT system dysregulations in these patients when considered as a whole.

Confirming our previous results [4], MDDs with current SBD exhibit HPT axis abnormalities evoking a form of central hypothyroidism consistent with a decreased hypothalamic TRH drive leading to a paucity of releasable TSH (particularly evident in the evening when the TSH circadian pattern approaches its acrophase) and a reduction in the activation of the thyroid gland. This hypothesis has already been discussed (see references [4,18] for an extended discussion). In the present study, our new finding is the unexpected increase in ACTH and GH responses to APO in some current SBDs, especially in recent suicide attempters.

Previous studies measuring the GH response to APO in depressed inpatients with a history of suicide attempts are scarce: blunted [6,12] or normal [14] responses have been reported in small patient samples. Besides differences regarding clinical features (such as the presence of anhedonia, motor retardation, melancholia or the time elapsed since the last suicide attempt), these discrepant findings may also be related to the diversity of factors that influence the GH response to APO [33]. Among them, the dose of APO administered subcutaneously appears to be a major determinant. Pitchot et al. [6,12] used 0.5 mg of APO but we used 0.75 mg in this and our previous studies [5,14,34]. It is therefore possible that 0.75 mg of APO stimulates more GH secretion than 0.5 mg, and consequently could better demonstrate DA receptor hyperresponsiveness. On the other hand, the 0.75 mg dose is the minimum dose that produces a stimulation of ACTH [34,43,44]. In an earlier study [14], we found comparable APO-GH and APO-ACTH/COR responses among healthy controls and DSM-IV major depressed inpatients classified according to the presence or absence of a history of suicidal attempts. However, we did not assess suicidal behavior as defined by the proposed DSM-5 specifiers (i.e., “current” or “in early remission”) as in the present study.

Usually GH and ACTH/COR responses to APO are related (albeit not always significantly in the present study), suggesting common physiological mechanisms [5]. GH and ACTH responses to APO are triggered via stimulation of GHRH and CRH, respectively, and though the role of D2 receptors at the pituitary level remains to be elucidated, they probably inhibit GH and ACTH secretion [45,46]. Hence, increased GH and ACTH responses in some current SBDs suggest increased D1 and/or D2 receptor responsiveness in the hypothalamus. Surprisingly, APO-COR responses are comparable across SBD and HC groups, whereas one would have expected that these responses would also be increased in current SBDs. One may therefore hypothesize that ACTH receptors in the cortex of the adrenal gland could be relatively hyposensitive to endogenous ACTH in some current SBDs. This is further supported by the fact that in current SBDs: (1) ∆ACTH and ∆COR values are not significantly correlated and (2) ACTHBL values are slightly higher (but not significantly) than in HCs and SBDs in early remission. However, since the HPA axis activity—as assessed with the DST—does not differ across the three diagnostic groups, this phenomenon remains unexplained at present.

The pathophysiological mechanisms underlying the increased sensitivity of hypothalamic DA receptors are not fully understood. Two main hypotheses can be considered. On the one hand, DA receptor hypersensitivity may be secondary to chronic decreased presynaptic DA output. This homeostatic change may be induced to restore baseline levels of DA receptor stimulation. Several lines of evidence suggest that D1-like and D2-like receptors are upregulated in the striatum and in the DA mesolimbic pathway after DA depletion [47,48]. In suicide attempters, reduced CSF-HVA (reflecting decreased DA turnover in the CNS; see reference [8] for a review) and increased D1 and D2/3 receptor function in the mesolimbic DA pathway have been reported, although not all studies agree (see reference [49] for a review). On the other hand, assuming that tonic DA stimulation can sensitize the entire DA system to phasic stimulation [50], hypersensitivity of DA receptors might reflect an increase in tonic DA tone. In other words, if the DA tone is increased, the response of sensitized neurons to DA, or a DA agonist such as APO, is increased too. This hypothesis is supported by findings from suicidal victims, where, regardless of the violence of the method, cortical HVA concentrations were higher than controls [9]. Given that 5-HT inhibits DA activity when this system is in hyperfunction, 5-HT hypofunction, frequently associated with suicidal behavior [1,26], could fail to dampen hyper DAergic activity, which could lead to more impulsive behaviors [1]. Further research is needed to investigate these hypotheses, especially in impulsive suicide attempters.

In the light of our results, different biological stages may be described among SBDs as a function of the time elapsed since the last suicide attempt. Current SBDs with a history of a recent attempt (≤2 months prior to evaluation) are characterized by both alterations of the DA and HPT systems (as reflected by increased ∆GH values associated with reduced ∆∆TSH values). Current SBDs with a history of past suicide attempt (between 3 and 11 months prior to evaluation) are characterized by alteration of the HPT axis without alteration of the hypothalamic DA function (as reflected by reduced ∆∆TSH values associated with normal ∆GH values). Since ∆∆TSH values are decreased in both recent and past suicide attempters, the reduced activity of the HPT axis does not appear to be a consequence of the suicide attempt itself but is likely a pre-existing state before the attempt [4]. It can therefore be hypothesized that a dysfunction of the DA system combined with decreased TRH drive could be a contributing factor to suicidality. Subsequently, the involvement of the mesocorticolimbic DA system in suicidality seems to gradually decrease with time, as indicated by negative correlations between the time elapsed since the last suicide attempt and ∆GH and ∆ACTH values. A rather similar phenomenon can be noted for the HPT axis, but with a certain delay compared to DA activity.

In MDDs with SBD in early remission (the last suicide attempt having occurred between 12 and 24 months before assessment), unaltered responses to APO and TRH tests suggest that the activity of both the DA and HPT systems are normal. Thus, despite hypothalamic DA system efficiency, the TRH-mediated compensatory mechanism (i.e., increased TRH secretion in response to decreased 5-HT activity [25]) is not triggered. In other words, unlike what is presumed for non-suicidal depressive patients [5], the integrity of the DA system does not elicit the emergence of the TRH homeostatic process in MDDs with SBD. However, the normality of both TRH and DA functions in SBDs in early remission might reflect a recovery process, counteracting the vulnerability to suicidal behavior linked to a persistent decrease in 5-HT function.

Some shortcomings of this study are worth discussing. First, we studied a small sample of SBDs, which may have reduced the statistical power of our analyses. By using non-parametric statistical methods, we adopted a conservative approach to analyzing the data. However, there were consistencies between the patterns of hormonal responses and SBD subgroups (regardless of the statistical tests used to analyze data), suggesting some validity in our results. Nevertheless, our findings must be considered preliminary until replicated in a larger patient population. Second, given the low number of violent suicide attempters, our results, which concern only unsuccessful suicide attempters, cannot be generalized to violent suicidal behavior. Third, among the confounding factors in the assessment of neurotransmitter function, insufficient washout period could be a major bias. However, our exclusion criteria and the length of the wash-out period seem sufficient to avoid drug-induced biases for the systems studied [34]. Fourth, we did not measure serum APO and DEX; thus, we cannot exclude pharmacokinetic differences as a source of bias in our results. Be that as it may, serum DEX and APO levels are unlikely to explain the differences in outcomes in depressed patients [5]. Finally, APO also has affinity for serotonin receptors (5-HT1A, 5-HT2A, 5-HT2B, and 5-HT2C) and α-adrenergic receptors (α1B, α1D, α2A, α2B, and α2C) [51]. Most of them are involved, to varying degrees, in the regulation of CRH and GHRH activity (see reference [52] for a review). Consequently, the changes in APO-induced GH and ACTH stimulation might also partly reflect 5-HT and α-adrenergic receptor sensitivity, although this hypothesis needs further investigation in MDDs with SBD.

## 5. Conclusions

The present study confirms that, regardless of the level of hypothalamic DA activity, the homeostatic TRH compensatory mechanism is not triggered in non-psychotic major depressed inpatients with suicidal behavior disorder (SBD), and this defect could play a role in the sustained deficit in 5-HT function consistently associated with suicidal behavior. We have extended the results of our previous work by finding that different levels of DAergic function and HPT axis activity are related to the course of illness in SBDs. Our data support the hypothesis that the co-occurrence of alterations in the DA and HPT systems (i.e., DA receptor hyperresponsiveness associated with decreased hypothalamic TRH drive) could contribute to the pathophysiology of suicidal behavior. Conversely, the normalization of TRH and DA functions might reflect a recovery process from suicidality. Our findings are also in line with promising therapeutic alternatives since the anti-suicidal effects of TRH [53] and ketamine [54]—a non-competitive antagonist at the N-methyl-d-aspartate receptor that stimulates the central secretion of TRH [55] and DA [56]—are under evaluation. Additionally, since no TRH analogs has been approved to date for the treatment of depression, the use of DA agonist agents—via stimulation of the TRH-compensatory mechanism—could represent a new therapeutic approach for suicide prevention [5]. Following this logic, one may advocate extending the indication of DAergic-based pharmacotherapies to individuals suffering from stress-related disorders (who present a high risk of suicidal behavior) such as depression [57] and post-traumatic stress disorder [58]. Further investigation of the specific effects of TRH and DA on suicidality is warranted.

## Figures and Tables

**Figure 1 brainsci-12-00621-f001:**
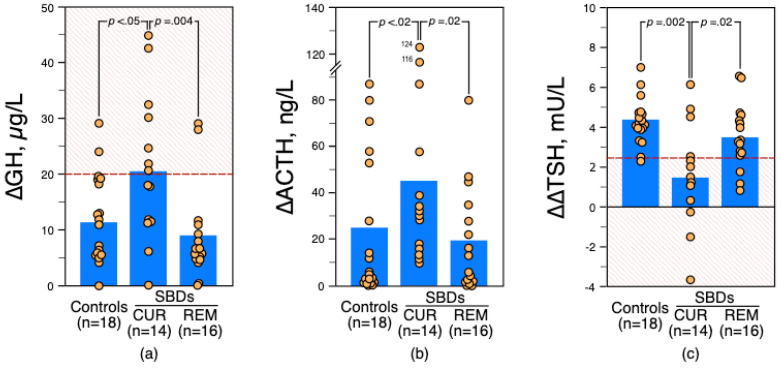
(**a**) Growth hormone (∆GH) and (**b**) adrenocorticotropic hormone (∆ACTH) responses to apomorphine (0.75 mg SC). (**c**) Difference between 11 PM and 8 AM maximum increments in thyrotropin (∆∆TSH) after 200 µg of protirelin given intravenously to healthy controls and depressed patients with current (CUR) or in early remission (REM) suicidal behavior disorder (SBD). Values are plotted individually; histograms represent the group mean. Comparisons were tested with a two-tailed *U*-test. The dashed lines indicate the threshold values that separate increased and normal ∆GH values and reduced and normal ∆∆TSH values (abnormal values are located in the shaded areas).

**Figure 2 brainsci-12-00621-f002:**
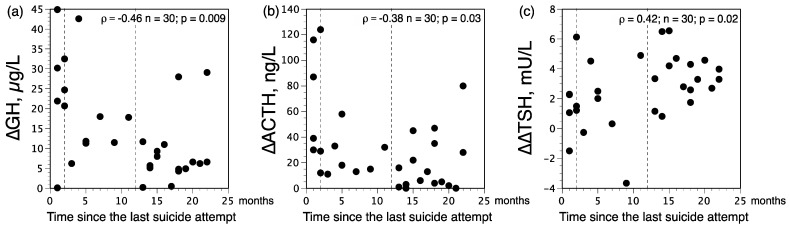
Scatterplots of (**a**) growth hormone (∆GH) and (**b**) adrenocorticotropic hormone (∆ACTH) responses to apomorphine (0.75 mg SC), and (**c**) difference between 11 PM and 8 AM maximum increments in thyrotropin (∆∆TSH) after 200 µg of protirelin given intravenously, against the time elapsed since the last suicide attempt (in months) in major depressed patients with suicidal behavior disorder (SBD). Values are plotted individually; *p* values derive from Spearman’s rank correlation analyses (ρ).

**Table 1 brainsci-12-00621-t001:** Demographic characteristics and biological data for normal controls and depressed patients with DSM-5 suicidal behavior disorder classified according to specifiers: current or in early remission.

	Control Subjects (*n* = 18)	SBD-REM (*n* = 16)	SBD-CUR (*n* = 14)
Age, years ^a^	37.8 ± 7.8	36.9 ± 11.4	33.0 ± 6.9
Gender	9 M/9 F	5 M/11 F	7 M/7 F
HAM-D	…	26.4 ± 4.5	28.0 ± 4.3
Melancholic features (*n*)	…	4	4
TSA, months	…	17.2 ± 3.1	3.9 ± 3.2 †††
NSA	…	1.4 ± 0.6	1.3 ± 0.6
**8 AM and 11 PM TRH tests**			
8 AM-FT4BL, pmol/L	13.6 ± 1.5	14.5 ± 3.0	12.3 ± 2.1 *
8 AM-FT3BL, pmol/L	5.1 ± 0.4	5.5 ± 0.7	5.3 ± 0.8
8 AM-TSHBL, mU/L	1.23 ± 0.44	1.42 ± 0.69	1.10 ± 0.38
8 AM-∆TSH, mU/L	9.32 ± 4.41	8.83 ± 3.61	7.30 ± 3.60
11 PM-TSHBL, mU/L	1.33 ± 0.75	1.12 ± 0.52	0.75 ± 0.29 *
11 PM-∆TSH, mU/L	13.66 ± 4.94	12.35 ± 3.67	9.00 ± 3.93 **†
∆∆TSH, mU/L	4.31 ± 1.16	3.54 ± 1.63	1.67 ± 2.55 ***†
**Apomorphine test**			
ACTHBL (ng/L)	23.0 ± 17.1	20.0 ± 8.6	30.9 ± 19.9
∆ACTH (ng/L)	24.8 ± 30.1	19.2 ± 22.7	44.1 ± 38.2 *†
CORBL (nmol/L)	282 ± 110	321 ±114	251 ± 77
∆COR (nmol/L)	176 ± 181	111 ± 131	167 ± 124
GHBL (µg/L)	0.6 ± 0.3	0.7 ± 0.3	0.7 ± 0.4
∆GH (µg/L)	12.0 ± 7.4	8.9 ± 8.3	21.0 ± 13.0 *†††
**Maximum post-DST**			
COR, nmol/L	44 ± 56	85 ± 103	66 ± 62

^a^ Values are mean ± SD. SBD, suicidal behavior disorder; CUR, current; REM, early remission; HAM-D, 17-item Hamilton Rating Scale for Depression; TSA, time elapsed between the last suicide attempt and the investigation; NSA, number of suicide attempts; FT4, free thyroxine; T3, free triiodothyronine; TSH, thyrotropin; ACTH, adrenocorticotropin hormone; COR, cortisol; GH, growth hormone; BL, basal concentration; ∆, peak concentration minus basal concentration; ∆∆TSH, 2300h-∆TSH minus 0800h-∆TSH; DST, dexamethasone suppression test. Comparisons between control and depressed groups, * *p* ≤ 0.05; ** *p*≤ 0.01; *** *p* ≤ 0.005; comparisons between SBD-REM and SBD-CUR groups, † *p* ≤ 0.05; ††† *p* ≤ 0.005 (by *U*-test).

**Table 2 brainsci-12-00621-t002:** Logistic regression analyses between presence (coded 1)/absence (coded 0) of current suicidal behavior disorder (SBD) and apomorphine test responses and hypothalamic-pituitary-thyroid activity in 30 depressed inpatients with SBD.

	Regression Coefficients (Standard Errors); *p*	Odds Ratios	95% CI Low-High	Overall Model Fit χ^2^; df; *p*
∆ACTH	0.03 (0.02); 0.065	1.03	0.99–1.06	4.92; 1; 0.027
∆COR	0.004 (0.003); 0.23	1.00	0.99–1.01	1.53; 1; 0.22
∆GH	0.11 (0.05); 0.017	1.12	1.02–1.22	8.63; 1; 0.003
Normal/High ^1^	1.94 (0.92); 0.035	7.00	1.14–44.97	5.19; 1; 0.023
FT4BL	−0.32 (0.16); 0.048	0.72	0.52–0.99	4.83; 1; 0.028
8 AM-∆TSH	−0.13 (0.11); 0.25	0.88	0.71–1.09	1.41; 1; 0.23
11 PM-∆TSH	−024 (0.11); 0.033	0.79	0.63–0.98	5.55; 1; 0.019
∆∆TSH	−0.46 (0.22); 0.04	0.63	0.40–0.98	5.77; 1; 0.016
Normal/Low ^2^	2.77 (0.91); 0.0025	15.89	2.65–95.21	11.46; 1; 0.0007

^1^ ∆GH: 0 = normal (≤20 µg/L), 1 = high (>20 µg/L). ^2^ ∆∆TSH: 0 = normal (>2.5 mU/L), 1 = low (≤2.5 mU/L). CI: confidence intervals.

**Table 3 brainsci-12-00621-t003:** Relationships among TRH, apomorphine and dexamethasone suppression tests in healthy controls and depressed patients with current (CUR) or in early remission (REM) suicidal behavior disorder (SBD).

	11PM-∆TSH	∆∆TSH	∆ACTH	∆COR	∆GH	COR Post-DST
**8 AM-∆TSH**						
Controls	0.93 ***	0.29	0.10	0.22	−0.04	−0.34
SBDs-CUR	0.83 ***	0.07	0.35	0.03	0.34	0.16
SBDs-REM	0.90 ***	0.14	−0.14	0.14	−0.15	−0.39
**11 PM-∆TSH**						
Controls		0.53 *	0.02	0.14	−0.09	−0.09
SBDs-CUR		0.54 *	0.30	0.17	0.41	−0.11
SBDs-REM		0.22	−0.24	0.03	−0.07	−0.34
**∆∆TSH**						
Controls			0.09	0.11	−0.05	−0.10
SBDs-CUR			−0.07	−0.01	0.28	−0.07
SBDs-REM			0.01	0.07	0.34	0.01
**∆ACTH**						
Controls				0.72 ***	0.46 *	−0.14
SBDs-CUR				0.48	0.51	0.09
SBDs-REM				0.69 **	0.45	−0.10
**∆COR**						
Controls					0.41	0.12
SBDs-CUR					0.35	0.04
SBDs-REM					0.42	−0.11
**∆GH**						
Controls						−0.32
SBDs-CUR						−0.07
SBDs-REM						−0.18

TSH, thyrotropin; ACTH, adrenocorticotrophic hormone; COR, cortisol; GH, growth hormone; DST, dexamethasone suppression test; ∆, increment after stimulation test; ∆∆TSH, difference between 11PM-∆TSH and 8AM-∆TSH. Spearman’s rank coefficient (ρ): * *p* < 0.05, ** *p* < 0.01, *** *p* < 0.001.

## Data Availability

The data presented in this study are openly available in https://www.data.gouv.fr/fr/datasets/r/54d20561-1c9d-4672-a36f-1108096f54fd (accessed on 22 March 2022).

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
