# Peer review of "Dopamine Function and Hypothalamic-Pituitary-Thyroid Axis Activity in Major Depressed Patients with Suicidal Behavior"

_brainsci, 2022, doi:10.3390/brainsci12050621_

Round 1
Reviewer 1 Report
This is a very interesting paper focused on the assessment of the dopamine function and hypothalamic-pituitary-thyroid axis activity in patients with major depression and suicidal behavior. The paper is well-written and is of interest for the journal. However, several minor changes should be made before publishing it.
In the abstract section, it would be useful to add a sentence focused on future perspectives. The paper is opening new avenues on the filed. In the abstract, it should be also reported.
In the introduction section, the authors are starting by a mention on the biological basis of suicidal behaviors in major depression. As a second paragraph, it would be useful to clarify the interrelationship between all these systems (5-HT, NA, HPA, etc).
What are the hypotheses of the study? They mentioned that the main aim was to clarify the relationship between the DA and HPT systems in depressed patients who commited suicide. The direction of the association can be postulated/hypothesized before explaining the methods.
In the methods section, data analysis and statistical analyses should be separated.
Section 2 is methods section and section 3 results.
The authors should revise and renumber all sections and subsections.
What are the clinical consequences of such results? The authors concluded that irrespective of the level of DA hypothalamic activity, other functions should be explored. What are the treatment implications? Future perspectives should be mentioned at the end of the section.
Author Response
please see the attached file, thank you

Reviewer 2 Report
In a confirmatory follow-up study, Duval et al. report that non-psychotic Major Depressive Disorder (MDD diagnosed per DSM-5) patients with suicidal behavior disorder (SBD) exhibited increased adrenocorticotropic hormone (ACTH) and growth hormone (GH) responses after dopamine receptor agonist (APO) administration, which is also a potent antiparkinsonian drug. Moreover, Duval et al. show that MDDs with SBD exhibited hypothalamic-pituitary-thyroid axis dysregulation.
Overall, the quality of the manuscript is excellent. Authors provide a comprehensive, well-referenced introduction and fully justify the experimental design and methods used in this retrospective study.
The main weakness is the sample size of the study and the resultant reduced statistical power. However, authors address this limitation in the discussion section.
Specific suggestions:
- Could authors insert a table with acronym definitions? It will be useful for the broad audience of the journal while reading the manuscript.
- Could authors justify the statistical tests employed in data analysis (Mann-Whitney U test; Spearman's rho statistics)? Were there any attempts to validate the results using alternative methods?
Author Response
please see the attached file, thank you

Reviewer 3 Report
In their manuscript, Duval and colleagues assessed the dopamine function and Hypothalamic-Pituitary-Thyroid axis
activity in major depressed patients with suicidal behavior. The manuscript is well-oganized and well-written. The prelimenary results provided by the Authors are potentially interesting. I have only one minor comment that can improve the quality of the manuscript.
-The Authors need to strengthen the concept that drugs targeting the dopaminergic system may be effective drugs for the treatment of individuals suffering from stress-related disorders (high risk of suicidal behavior) such as depression and PTSD. In this respect, I suggest to discuss and add the following references ( PMID: 31057408; PMID: 23904408)
Author Response
please see the attached file, thank you
